# No increased circular inference in adults with high levels of autistic traits or autism

**Nikitas Angeletos Chrysaitis**[1], **Renaud Jardri**[2,3], **Sophie Denève**[2‡], **Peggy Seriès**[1]*

**1** Institute for Adaptive and Neural Computation, School of Informatics, University of Edinburgh, Edinburgh, United Kingdom, **2** École Normale Supérieure, Institut de Sciences Cognitives, LNC (INSERM U960), Paris, France, **3** Université de Lille, INSERM U1172, Lille Neuroscience & Cognition Research Centre, Plasticity & SubjectivitY (PSY) team, CHU Lille, Lille, France

‡ Unavailable
* pseries@inf.ed.ac.uk

**Data Availability Statement:** Experiment code, models, and anonymized data can be found at https://osf.io/yqug2/.

## Abstract

Autism spectrum disorders have been proposed to arise from impairments in the probabilistic integration of prior knowledge with sensory inputs. Circular inference is one such possible impairment, in which excitation-to-inhibition imbalances in the cerebral cortex cause the reverberation and amplification of prior beliefs and sensory information. Recent empirical work has associated circular inference with the clinical dimensions of schizophrenia. Inhibition impairments have also been observed in autism, suggesting that signal reverberation might be present in that condition as well. In this study, we collected data from 21 participants with self-reported diagnoses of autism spectrum disorders and 155 participants with a broad range of autistic traits in an online probabilistic decision-making task (the fisher task). We used previously established Bayesian models to investigate possible associations between autistic traits or autism and circular inference. There was no correlation between prior or likelihood reverberation and autistic traits across the whole sample. Similarly, no differences in any of the circular inference model parameters were found between autistic participants and those with no diagnosis. Furthermore, participants incorporated information from both priors and likelihoods in their decisions, with no relationship between their weights and psychiatric traits, contrary to what common theories for both autism and schizophrenia would suggest. These findings suggest that there is no increased signal reverberation in autism, despite the known presence of excitation-to-inhibition imbalances. They can be used to further contrast and refine the Bayesian theories of schizophrenia and autism, revealing a divergence in the computational mechanisms underlying the two conditions.

## Author summary

Perception results from the combination of our sensory inputs with our brain's previous knowledge of the environment. This is usually described as a process of *Bayesian inference* or *predictive coding* and is thought to underly a multitude of cognitive modalities. Impairments in this process are thought to explain various psychiatric disorders, in particular autism and schizophrenia, for which similar Bayesian theories have been proposed despite

**Funding:** N.A.C. was funded by the UKRI Centre for Doctoral Training in Biomedical Artificial Intelligence of the University of Edinburgh (EP/S02431X/1). The funders had no role in study design, data collection and analysis, decision to publish, or preparation of the manuscript.

**Competing interests:** The authors have declared that no competing interests exist. Author Sophie Denève was unable to confirm their authorship contributions. On their behalf, the corresponding author has reported their contributions to the best of their knowledge.

differences in their symptoms. Recently, a new model of Bayesian impairment in schizophrenia was proposed and validated using behavioural experiments, called the 'circular inference' model. In the current study, we used the same task and computational modelling to explore whether circular inference could also account for autism spectrum disorders. We find that participants with autistic traits or self-reported diagnoses of autism do not present increased levels of circularity. This is the first study to investigate circular inference in autism, and one of the very few to explore possible autism and schizophrenia impairments with the same task and identical analytical methods. Our findings indicate one potential way in which the explanations of the two conditions might differ.

## Introduction

Autism spectrum disorder (ASD) and schizophrenia (SCZ) are two heterogeneous mental disorders with a complicated relationship [1,2]. While the term 'autism' was initially used to refer to one of schizophrenia's symptoms [3], the two disorders have since been considered as separate conditions and have been studied as such by most researchers. Despite that, numerous links have been observed between them, from behavioural and neurophysiological similarities in social cognition impairments [4,5], to immune [6] or intestinal [7] dysregulation and genetic overlap [8], among others. Such findings suggest that the relationship between schizophrenia and ASD should be more thoroughly explored, within a framework that is able to handle and explain their differences [9,10].

In Bayesian theories of perception and cognition, the brain is viewed as constantly making probabilistic calculations in order to infer the true state of the environment. The information coming from sensory inputs is captured by the likelihood function and is combined with prior beliefs about the environment, in a process akin to Bayesian inference [11]. This framework has been widely adopted in both ASD and SCZ research, with a frequently proposed hypothesis for both disorders being that sensory inputs are overweighted relative to prior beliefs [12–16] (see [17–19] for an alternative SCZ hypothesis). In schizophrenia, this theory attempts to explain the tendency of patients to jump to conclusions [20] and their partial immunity to perceptual illusions [21], with hallucinations and delusions being interpreted as the formation of bizarre beliefs to account for strange, hypersalient sensory data [22]. Intriguingly, the hypothesis of overweighted sensory information is also suggested to account for most of ASD's symptoms, such as sociocognitive impairments, attention to detail, sensory hypersensitivity, and decreased susceptibility to illusions [15]. The similarity of the proposed theories for autism and schizophrenia is surprising given their distinct symptomatology. However, very few Bayesian studies have examined both conditions using the same experimental or computational paradigm, which would be crucial for understanding their relationship and mechanisms of action.

In 2013, Jardri and Denève proposed a new computational explanation for schizophrenia, called *Circular Inference* [23], motivated by an attempt to understand the potential consequences of the increased excitation-to-inhibition (E/I) ratio that is associated with the condition [24,25]. Using hierarchical network simulations, they showed that inhibitory impairments in the cortex might lead to sensory evidence or prior beliefs being reverberated throughout the network that the brain uses to represent the environment, overwhelming the inferential process. Sensory input reverberation could cause the reported 'jumping to conclusions' bias in schizophrenia, where patients get overconfident in their beliefs based on relatively little evidence [26]. The positive symptoms, then, can be seen as an extension of the same process,

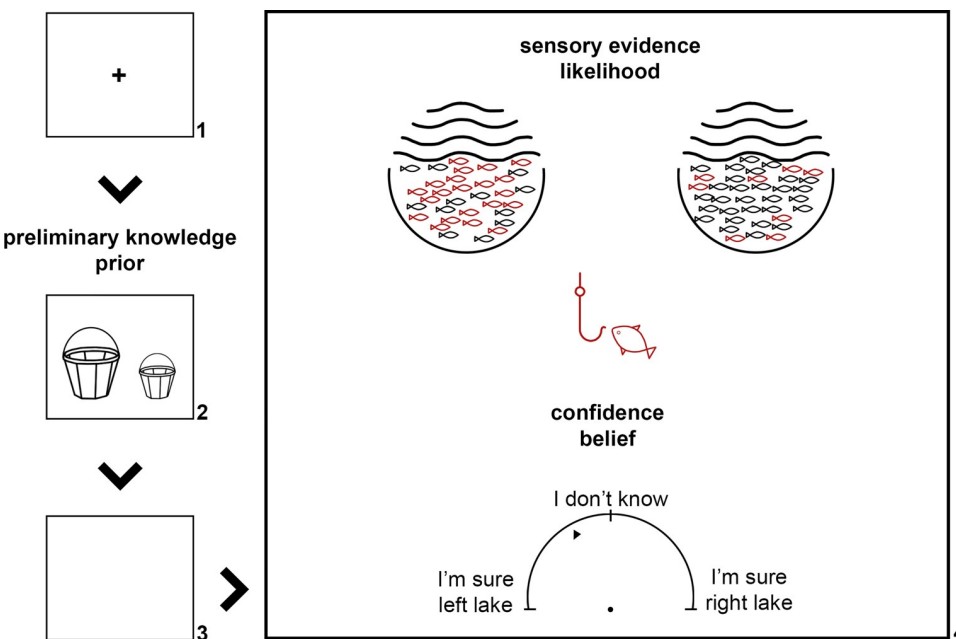

**Fig 1. An outline of the four stages of the fisher task.** *1) The fixation cross is presented; 2) participants are shown the preference of the fisherman, visualised as two baskets of varying sizes, one for each lake; 3) a blank screen is presented; 4) participants are shown the fish proportions and are asked to make a confidence estimate about the lake of origin of the fish (Adapted from Jardri et al., 2017 [27]).*

where hallucinations and delusions are produced by misplaced certainty in noisy perceptual and other non-sensory information, respectively.

Jardri et al. supported this hypothesis with behavioural evidence from 25 SCZ patients and 25 controls [27], using a probabilistic variant of the beads task [28], called the *fisher task*. In the fisher task, subjects are asked to estimate the chance that a red fish caught by a fisherman came from one of two lakes, while being presented with the lake preferences of the fisherman and the proportions of red fish in each lake (Fig 1). The researchers interpreted the preferences (which were presented first) as a Bayesian prior and the fish proportions as the sensory evidence. They showed that all participants exhibited signs of signal reverberation. Importantly, they found that sensory evidence was reverberated more in patients, with the magnitude of reverberation being correlated with their positive symptoms. A following study confirmed these findings, utilising a social version of the beads task in a sample of 35 patients with schizophrenia or schizoaffective disorder and 40 controls [29]. The researchers found that the circular inference model fitted best the participants' behaviour, with increased sensory reverberation in patients. They also presented strong evidence for an association between that reverberation and various clinical features in patients (e.g., delusions, anhedonia-asociality).

Impaired inhibition has been strongly associated with autism [30–34]. A question that arises naturally is, therefore, whether circular inferences are present in ASD, and whether they would then be of the same nature as in schizophrenia (e.g., sensory vs prior reverberation). In the present study, we aimed to assess cue integration across a sample with a broad range of autistic traits, which also included some autistic participants ('autistic' is the preferred term by people on the autism spectrum [35]). This allowed us to investigate signal reverberation within a dimensional as well as a more traditional, categorical view of autism [36–39]. To achieve that, we utilised an online version of the fisher task, and both circular inference and more traditional Bayesian models. This provided us with an opportunity to explore the influences of

ASD in probabilistic decision-making, while also allowing an additional, qualitative comparison with past SCZ findings.

## Methods and materials

### Ethics statement

The present study was approved by the University of Edinburgh, School of Informatics Ethics committee (RT number 29368).

### Sample

We recruited 204 naive adults; 61 voluntarily via our social media networks and 143 with fixed monetary compensation via the Prolific recruiting platform [40]. All participants had normal or corrected-to-normal vision and were not taking any psychotropic medication. 28 subjects were excluded for providing low quality data (Section A3 in S1 Supplementary Information). The final sample included 102 male and 71 female participants, with a median age of 26.6 years. The study was conducted online. Participants were presented with detailed information about the study and had to click a button to indicate consent for the experiment to start.

Half of the Prolific subsample was selected to have a self-reported diagnosis of ASD or to identify as part of the autism spectrum (Section A1 in S1 Supplementary Information), with 21 subjects having a diagnosis in the final sample. All participants filled in the Autism Spectrum Quotient (AQ) questionnaire [41] and the 21-item Peters et al. Delusions Inventory (PDI) [42]. The final sample showed indeed stronger autistic traits (mean 22.9, SD 6.5) than what is usually found in the general population (mean 16.9, SD 5.6) [43], but no difference in delusional ideation (mean 6.1, SD 3.1 vs mean 6.7, SD 4.4) [42]. Interestingly, the participants with the ASD diagnoses had AQ scores on the low-end (mean 28.0, SD 8.0) compared to those reported in the literature for autistic individuals (mean 35.2, SD 6.3) [43]. Statistical power for our tests could not be calculated, as model parameters were not verifiably following any known distribution. However, the strength of Jardri et al.'s findings [27] suggests that comparable effects would reach statistical significance in our larger sample, according to an exploratory analysis (Section A4 in S1 Supplementary Information).

### Procedure

The task was kept as similar to the original fisher task [27] as possible. The participants were shown a fisherman having caught a red fish and were asked which of two lakes the fish was caught from. To make this decision, they were presented with two kinds of information in each trial: 1) the preferences of the fisherman for each of the lakes, visualised as two baskets of varying sizes (prior); 2) the proportions of red versus black fish in each lake, visualised as 100 fish in two lake drawings (sensory evidence or likelihood). Subjects were instructed to gauge their confidence and respond using a continuous semi-circular scale, ranging from 'I'm sure LEFT LAKE' to 'I'm sure RIGHT LAKE', with 'I don't know' in the middle. Confidence estimates were interpreted probabilistically, in a continuous manner, with a click on the left edge of the scale corresponding to a probability of 1 for the fish originating in the left lake (0 for the right) and vice versa.

Trials were structured as follows (Fig 1): Initially, a fixation cross was shown for 800ms, followed by the two baskets for 1000ms, and a blank screen lasting 50ms. Then, the lake drawings, the fisherman with the red fish, and the scale appeared on the screen until the subject gave a response. Participants were presented with detailed instructions which they could view many times before proceeding to the task. The instructions made clear that participants should

respond 'as fast and as accurately as possible'. After the instructions, subjects completed 11 training trials with easy stimulus combinations to acclimate themselves with the task.

Due to concerns about participants' potential distractibility in an online environment if the task was too long, we reduced the number of trials to 130 (Section A2 in S1 Supplementary Information). The trials appeared in a random order, with lake drawings being different for every trial. Every 22 trials, the participants were prompted to take a break, which they could end with the press of a button. Lakes had 9 possible ratios of red to black fish, while baskets appeared in 9 possible sizes, both corresponding to the probabilities 0.1 to 0.9. In all trials, likelihoods and priors were complimentary (e.g., if the left fish proportions were 0.3, the right would be 0.7). Therefore, probabilities mentioned in the text refer to the left lake, as the probabilities for the right can be immediately inferred.

## Model-free analysis

Linear mixed-effects models (LMEs) were used to verify that participants combined the information of both baskets and fish ratios when making their decisions and to investigate any possible interactions with autistic traits. We chose the absolute confidence of the participants as the response variable ($|c- 0.5|$, with c being the participant confidence estimate). We modelled the following as fixed effects with repeated measures across the subjects in all LMEs: i) the absolute likelihood ($|likelihood- 0.5|$); ii) the prior congruency, that is how much the prior agreed with the likelihood ($|prior- 0.5| * sgn[(prior- 0.5)(likelihood- 0.5)]$); iii) the reaction times, which were used to investigate the possibility of a speed-accuracy trade-off. All LMEs also included the two-way interaction between i and ii, with the participants being treated as a random factor. We analysed our results with 5 different LME variants. The first one, LME_core only used the aforementioned components. LME_AQ expanded upon LME_core by including a fixed effect for AQ and the two- and three-way interactions of AQ with i and ii. LME_PDI was the same as LME_AQ but with the PDI scores instead of the AQ. Then, LME_full, used both AQ and PDI and their interactions with i and ii, but no interactions between them. Finally, LME_rtInteract expanded upon the LME_full to include interactions between AQ or PDI and reaction times. Full specification of the models in Wilkinson notation can be found in Section B1 of S1 Supplementary Information.

## Bayesian models

Data were fitted with four Bayesian models: Simple Bayes (SB), Weighted Bayes (WB), and two variants of the circular inference model: Circular Inference–Interference (CII) and Circular Inference–No Interference (CINI). Originally [23], the inferential processes expressed by these models were simulated in a hierarchical network, where priors corresponded to top-down signals and likelihoods to bottom-up ones. In the current study, we followed Jardri et al. in fitting simplified models, that capture the network effects with significantly fewer free parameters [27].

SB combines the two sources of information using Bayes' theorem. This is expressed in logits as

$$L_c = L_p + L_s, \tag{1}$$

with subscript *p* corresponding to trial prior, *s* to sensory evidence, and *c* to the confidence estimate, while *L* denotes the respective logit.

WB expands upon SB:

$$L_c = F(L_p, w_p) + F(L_s, w_s), \tag{2}$$

where $F$ is the sigmoid function

$$F(L, w) = \ln\left(\frac{we^L + 1 - w}{(1 - w)e^L + w}\right),$$

(3)

allowing for the underweighting of priors or likelihoods. Each weight $w$ determines the influence of the corresponding signal to the confidence estimate. This depends on how the reliability of that signal is estimated by each participant.

CII has the form:

$$L_c = F(L_p + I, w_p) + F(L_s + I, w_s),$$

(4)

$$I = F(a_p L_p, w_p) + F(a_s L_s, w_s),$$

(5)

where top-down and bottom-up signals get reverberated, interfering with one another, and end up corrupting prior beliefs and sensory evidence by the same amount, $I$. Parameters $a_p$ and $a_s$ affect the number of times the respective information is overcounted, expressing the signals' reverberation.

CINI is similar to CII, but it assumes that both signals get reverberated or overcounted separately and are only combined at the end of the process:

$$L_c = F(L_p + F(a_p L_p, w_p), w_p) + F(L_s + F(a_s L_s, w_s), w_s).$$

(6)

SB has 0 free parameters, WB 2, and both CII and CINI have the same 4. True parameter ranges were [0.5, 1] for the weights ($w$) and [0, 60] for the reverberation parameters ($a$); however, these were rescaled to [0, 1] so that they could be easily compared with those reported by Jardri et al. 60 is an arbitrary upper limit, that however is high enough for our purposes, as no parameter approached it (max non-rescaled CINI $a$ = 29.02). In the rest of this article, we will be referring exclusively to the rescaled parameters, however the word 'rescaled' will be omitted for conciseness. A (rescaled) weight value of $w = 0$ shows no influence of the corresponding signal, while both $w = 1$ make WB equivalent to SB and both $a = 0$ make CII and CINI equivalent to WB. The difference between CII and CINI is subtle, but important. In CINI, the sensory and prior signals are combined linearly, while in CII, one signal's influence on the model estimate depends on the strength of the other, due to the interference between them. Fig 2 illustrates the behavioural patterns predicted by the different models.

We followed Jardri et al., assuming Gaussian noise in the logit model estimates ($L_c$), and therefore fitted models via least squares, which is equivalent to maximum likelihood estimation in that case. Model comparison was performed using an approximation of the Bayesian information criterion (BIC) for normally distributed errors,

$$\text{BIC} = n \ln(\sigma^2) + k \ln(n),$$

(7)

where $n$ is the number of datapoints, $k$ the number of free parameters, and $\sigma$ the model's mean squared error. To choose a model across all subjects, we followed the random-effects Bayesian model selection [44], implemented in the SPM12 [45]. Group-level BIC [46], a fixed-effects approach, produced similar results.

## Statistical analysis and validity of results

We investigated the hypothesis of an association between autism and circular inference ($H_1$) in three ways: 1) correlations between model parameters and total AQ scores; 2) differences between the low- and high-AQ groups, defined as participants in the top and bottom 15% of

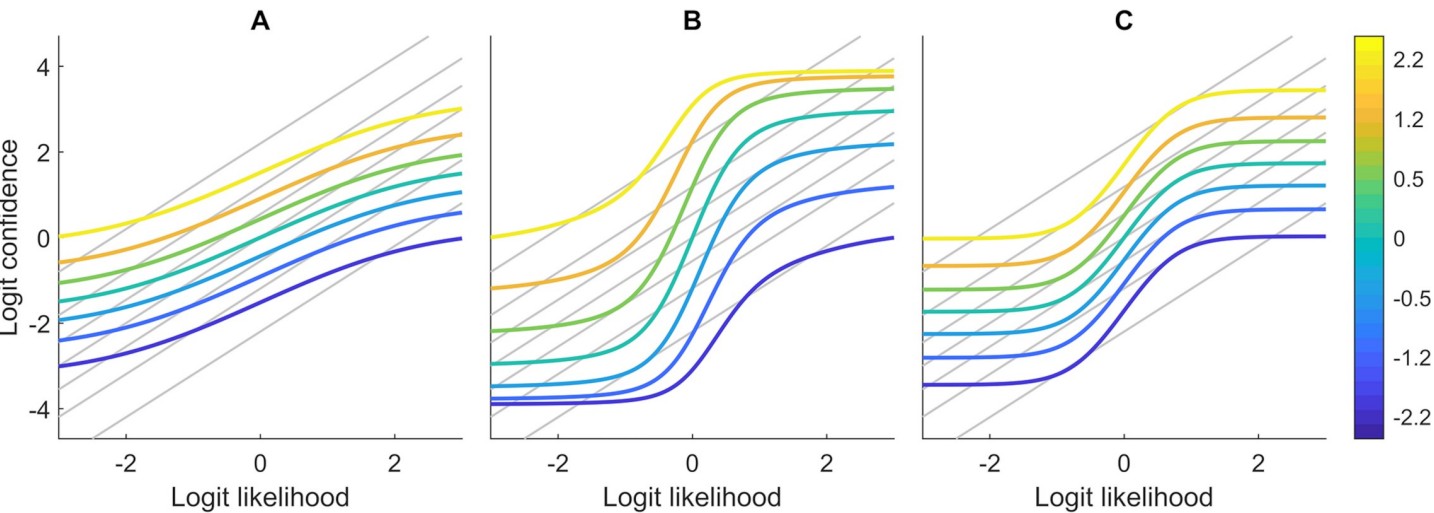

**Fig 2. Illustration of WB (A), CII (B), and CINI(C) behaviour.** The graphs show how logit model confidence estimates change as a function of logit likelihood (fish proportions). Different colours represent different prior values (basket size) and grey lines represent the SB model predictions. The SB model simply combines the information of the two signals by adding their logits. WB can underweight either or both signals, while, in addition to that, the circular inference models allow for signal overcounting. In CII, the contribution of the likelihood on the confidence estimate depends on the prior value and vice versa. In contrast with that, in the CINI model, each source of information affects the confidence independently, and therefore the graph lines are completely parallel to each other. Parameter values were the same for all models ($a_p$ = 0.02, $a_s$ = 0.05, $w_p$ = 0.8, $w_s$ = 0.06).

the sample (AQ $\geq$ 30, $n$ = 29, M/F: 13/14 vs AQ $\leq$ 16, $n$ = 30, M/F: 15/15); 3) differences between subjects with an ASD diagnosis and those without, who also did not identify as part of the autism spectrum (ASD, $n$ = 21, M/F: 13/8 vs ND, $n$ = 61, M/F: 39/22; answers 1, 2 vs 5 in Section A1 in S1 Supplementary Information). The nonparametric measures of Kendall rank correlation coefficient and Mann-Whitney U test were chosen, as model parameters were not normally distributed (*Shapiro-Wilk test*; $p \leq 0.0068$) and there is no reason to expect a linear relationship between them and psychiatric traits. The common language effect size statistic (*f*) was reported for the Mann-Whitney U [47]. All analyses were performed in MATLAB R2020a.

To quantify the evidence for the null hypothesis ($H_0$) in favour of the alternative one ($H_1$), we calculated the Bayes factors 01 ($BF_{01}$) for each of our tests. $1 < BF_{01} \leq 3$ constitutes weak evidence in favour of $H_0$, $3 < BF_{01} \leq 20$ positive evidence, and $BF_{01} > 20$ strong [48]. Note that $BF_{10} = 1/BF_{01}$. Bayes factors were calculated in JASP 0.14, using the default priors [49]. To verify the fitting and model selection processes, we performed parameter and model recovery on CINI and CII using the current set with the 130 trials, as SB and WB scored very poorly in model comparisons.

Both models showed moderate recovery for the reverberation parameters (CINI $a_p$, $r$ = 0.54; $a_s$, $r$ = 0.58; CII $a_p$, $r$ = 0.54; $a_s$, $r$ = 0.71), although this was partly due to Pearson's

**Table 1. Confusion matrix for model recovery.**

|  |  | Recovered | |
|---|---|---|---|
| **Simulated** |  | CINI | CII |
|  | CINI | 799 | 201 |
|  | CII | 185 | 815 |

Perfect model recovery would result in 1000 participants in the (CINI, CINI) and (CII, CII) cells, and 0 in the rest.

correlation sensitivity to outliers [50] (for details see Section B3 in S1 Supplementary Information). The models exhibited excellent recovery for the weight parameters (CINI $w_p$, $r = 0.96$; $w_s$, $r = 0.91$; CII $w_p$, $r = 0.94$; $w_s$, $r = 0.93$). They also showed no correlation between different parameters (Tables B3 and B4 in S1 Supplementary Information). Model recovery was good for both models, with approximately 80% of the simulated participants being better fitted by their generating model (Table 1).

## Results

### Model-free findings

Participant responses adapted to changes in both priors and likelihoods, showing that they took both sources of information into account to make their confidence estimate (Fig 3). Despite that, their behaviour was not strictly Bayesian. A change from 0.5 to 0.4 or 0.6 in either prior or likelihood corresponded to a disproportionally large shift in the average response, indicative of signal reverberation.

Among the linear mixed-effects models, the one which achieved the smallest BIC was LME_core (ΔBIC: LME_PDI, 17; LME_AQ, 35; LME_full, 51; LME_rtInteract, 69). All models confirmed the influence of both absolute likelihood (e.g., LME_core: $t = 44.50$, $p < 10^{-323}$) and prior congruency (e.g., LME_core: $t = 24.63$, $p = 10^{-132}$), as well as the interaction of the two components (e.g., LME_core: $t = 25.20$, $p = 10^{-138}$). Despite the LME_core being the best model, both LME_PDI and LME_full showed significant association between absolute confidence and non-clinical delusional beliefs (PDI) (e.g., LME_PDI results: $t = 2.08$, $p = 0.037$) and an interaction between absolute likelihood and PDI (e.g., LME_PDI results: $t = 2.31$, $p = 0.021$). However, neither the influence of autistic traits (AQ) or its interactions with model components were significant in the LME_AQ and LME_full models. Reaction times showed a negative relationship with absolute confidence in all models (e.g., LME_core: $t = -17.01$, $p = 10^{-64}$), which is presumably a result of participants taking more time to respond when they are uncertain [51]. Importantly though, the LME_rtInteract achieved the worst BIC score, with no interaction between psychiatric traits and reaction times (LME_rtInteract: PDI

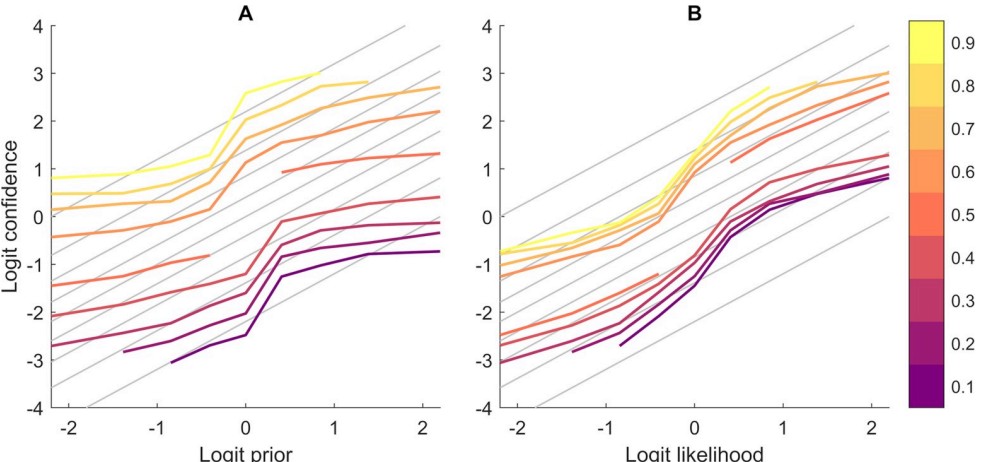

**Fig 3. Average logit confidence estimates for all participants as a function of priors (A) and likelihoods (B).** *Logit confidence estimates for the left lake increase following an increase in either prior probability for the left lake (baskets) or likelihood (fish ratios), showing that participants incorporate both information sources in their decision-making. However, their behaviour is far from strictly Bayesian, as evidenced by the differences between coloured and grey lines (SB confidence estimates). Different colours correspond to different likelihood (probability) values in the left graph and different prior (probability) values in the right.*

t = 1.29, p = 0.20; AQ t = 0.72, p = 0.47). This suggests that any possible relationship between AQ or PDI and participant behaviour is not a result of differences in time management. The full LME results can be found in Section B1 of S1 Supplementary Information.

## Model-based findings

Both random- and fixed-effects model comparisons showed that Circular Inference–No Interference was the best fitting model, followed by Circular Inference–Interference (Fig 4). Since model fit plots showed that both CINI and CII fit the data relatively well (Fig D1 in S1 Supplementary Information), for the sake of completeness, we conducted the same analysis with parameters from both models. Results from CINI are reported below, while those from CII can be found in S1 Supplementary Information (Section D2).

There was no evidence of a relationship between prior or likelihood reverberation and total AQ scores (Table 2). The only correlation that reached an (uncorrected) *p*-value of lower than 0.05 was a negative correlation between AQ and the CINI prior weight ($\tau = -0.12$, $p = 0.02$, $BF_{10} = 1.57$), but this did not survive adjusting for multiple comparisons [52]. Furthermore, the low- and the high-AQ groups behaved in a similar way (Fig 5), and the comparison between the parameters of high- and low-AQ groups did not reveal any difference, neither did the comparison between the ASD participants and those with no diagnosis (ND) (Fig 6 and Table 3). Since it is possible that ND subjects with high autistic traits have an undiagnosed autism spectrum disorder, we performed an additional comparison between the ASD group and the subgroup of ND participants with weak autistic traits ($AQ \leq 17$, $n = 21$, M/F: 13/8). No difference between these groups was found (Section D1 in S1 Supplementary Information). A weak positive correlation was found between PDI and the likelihood weight ($\tau = 0.13$, $p = 0.02$, $BF_{10} = 2.08$; Table 2), that again is not significant when corrected. No relationship was present between psychiatric traits or diagnoses and CII parameters (Section D2 in S1 Supplementary Information).

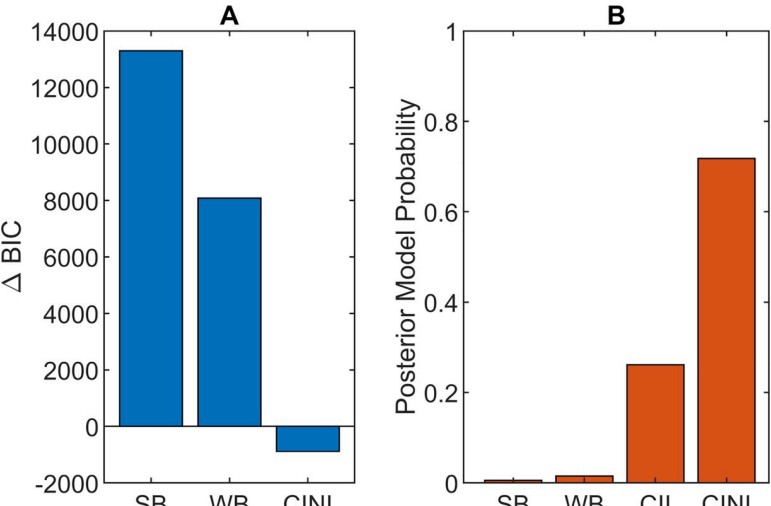

**Fig 4. Results of fixed (A) and random (B) model comparisons.** *(A) Group-level ΔBIC is defined as the sum of individual participant BIC scores for each model minus the sum for CII, used as a baseline, as it was the winning model in the Jardri et al. study* [27]. *The lower the BIC the better the model, with differences of more than 20 between BIC scores considered very strong evidence* [48]. *ΔBIC for CII is by definition 0. (B) Posterior model probabilities calculated using Bayesian model selection* [44]. *Both measures clearly show that Circular Inference models better account for the data, with CINI being a slightly better fit than CII.*

**Table 2. Kendall rank correlations between CINI parameters and psychiatric traits.**

| CINI params | AQ | | | PDI | | |
|---|---|---|---|---|---|---|
| | $\tau$ | $p$ | $BF_{01}$ | $T$ | $p$ | $BF_{01}$ |
| $a_p$ | 0.04 | 0.5 | 7.98 | −0.04 | 0.48 | 7.76 |
| $a_s$ | −0.02 | 0.65 | 9.11 | 0.01 | 0.85 | 9.94 |
| $w_p$ | −0.12 | 0.02 | 0.64 | 0.07 | 0.20 | 4.14 |
| $w_s$ | −0.02 | 0.69 | 9.35 | 0.13 | 0.02 | 0.48 |

Total AQ scores and Y/N PDI scores were used for the correlations. $\tau$ signifies the correlation coefficient. p-values are not adjusted for multiple comparisons. $BF_{01}$ stands for the Bayes factor 01, with higher values corresponding to stronger evidence for the null hypothesis.

## Discussion

In the present study, we investigated the relationship between circular inference and autistic traits or autism. Circular inference is an impairment in Bayesian hierarchical networks where top-down or bottom-up signals get reverberated throughout the network, becoming significantly amplified [23]. We hypothesised that stronger autistic traits and ASD diagnoses would be associated with stronger reverberation of priors or sensory evidence. We used the fisher task, a probabilistic decision-making task that had been used previously with patients with schizophrenia [27]. To our knowledge, this is the first study to explore signal reverberation in ASD. Our analysis showed that the circular inference models perform best across the whole sample, similarly to previous results [27,29]. However, our hypothesis was refuted. Specifically, no correlation was found between autistic traits and either reverberation parameter. Similarly, there were no differences in these parameters between the groups with the strongest and weakest autistic traits, and no differences between the autistic subjects and those with no self-reported diagnosis.

Circular inference attempts to model the effects of increased excitation-to-inhibition ratio, a phenomenon which has been strongly associated with schizophrenia [24,25]. Indeed, Jardri et al. found clear experimental evidence for stronger likelihood reverberation in SCZ patients, using the fisher task [27]. On that account, the absence of any difference between our participant groups is surprising, given the observed inhibitory impairments in ASD [33,34] and the commonalities between autism and schizophrenia regarding E/I imbalances [53,54]. Moreover, prominent computational explanations for the two conditions suggest similar Bayesian impairments between them. Specifically, it has been proposed that an imbalance of likelihoods to priors, in favour of the former, lies at the heart of both ASD and SCZ [12–16]. This seems to be contradicted by our findings, which showed no increase in reverberation along the autism spectrum, despite the presence of such an increase in schizophrenia. This is further exhibited in a qualitative comparison between the conditions, which showed higher likelihood reverberation in SCZ (Fig E1 in S1 Supplementary Information). A divergence in the Bayesian mechanisms of the two conditions has also been observed by Karvelis et al., which showed an association between autistic traits and increased sensory precision, but no discernible imbalance in schizotypy, in a statistical learning task [55]. A partial divergence was also found by Noel et al., in an audio-visual synchrony task, where patients with schizophrenia showed increased unreliability in sensory representations, in addition to differences in their priors, which they shared with the autistic participants [56]. No other studies are known to us that compare ASD and SCZ using the same tasks and Bayesian models, despite the commonalities between their computational explanations (for reviews see [14,15]).

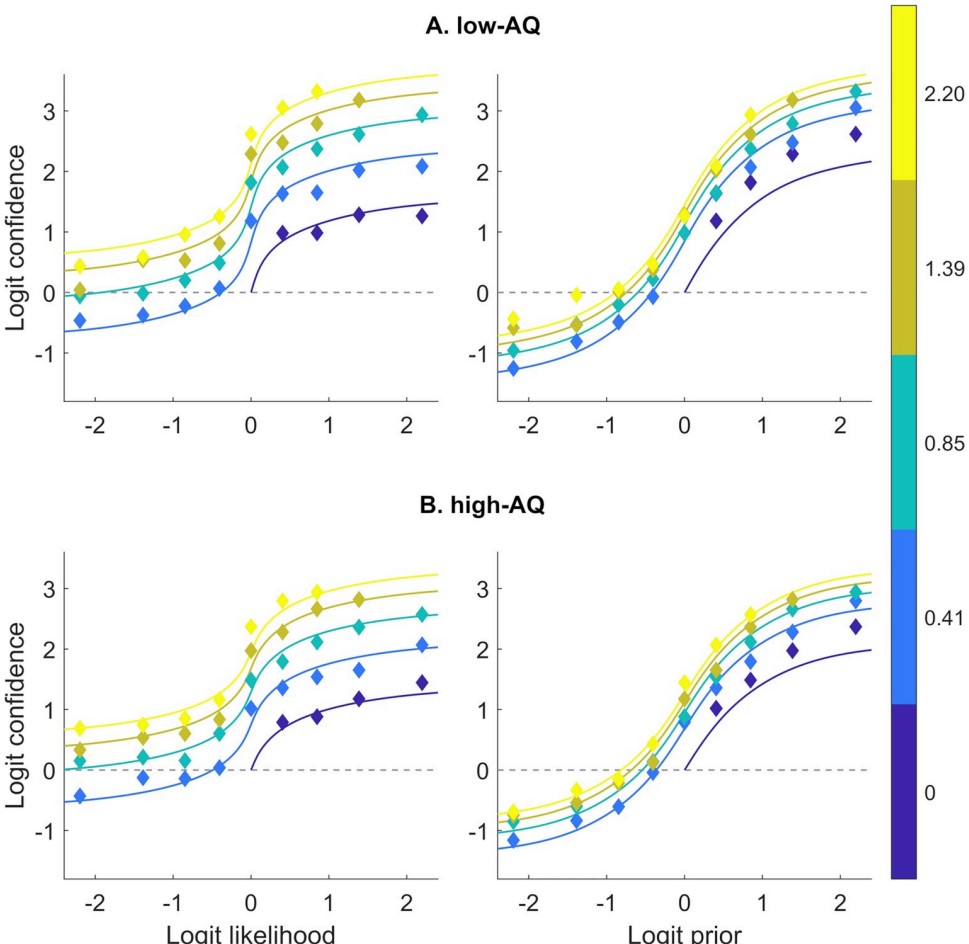

**Fig 5. Participant confidence estimates and CINI model fits for the low-AQ (A) and the high-AQ (B) groups.**
*Model and participant logit confidence as function of logit likelihoods and priors. Coloured lines represent model
predictions and rhombuses the participant confidence estimates. Different colours represent logit likelihood in A and logit
prior values in B and are equivalent to probabilities of 0.5 to 0.9. Since both the task and the CINI model structure are
symmetrical around 0 logit confidence (0.5 probability), participant estimates have been averaged between symmetric
trials to reduce noise (e.g., a trial with a logit prior of –1 and a logit likelihood of 2 is symmetrical to one with a logit prior
of 1 and a logit likelihood of –2).*

In agreement with the findings of Jardri et al. [27], we found compelling evidence for signal
reverberation across our sample. Interestingly though, one variant of the model, Circular
Inference–No Interference (CINI), was a better fit for our data compared to the other variant,
Circular Inference–Interference (CII), contrary to the Jardri et al. study. Additional analysis of
the Jardri et al. dataset revealed that this is partially because we used fewer trials than in the
original study (Table E3 in S1 Supplementary Information). Furthermore, even in the original
dataset, CII was dominant mostly in the SCZ subsample, while it performed equally well with
CINI in controls. The dominance of CINI across our sample (Table E3 in S1 Supplementary
Information) is an indication that prior beliefs and sensory evidence are reverberated, even in
healthy participants. While we can only speculate about the possible neurobiological underpin-
nings of our circular inference models, we proposed that reverberation arises due to an
increased E/I ratio, based on the network model of Jardri & Denève [23]. If that is the case,
CINI might correspond to a weak or localised E/I imbalance, affecting the signals only

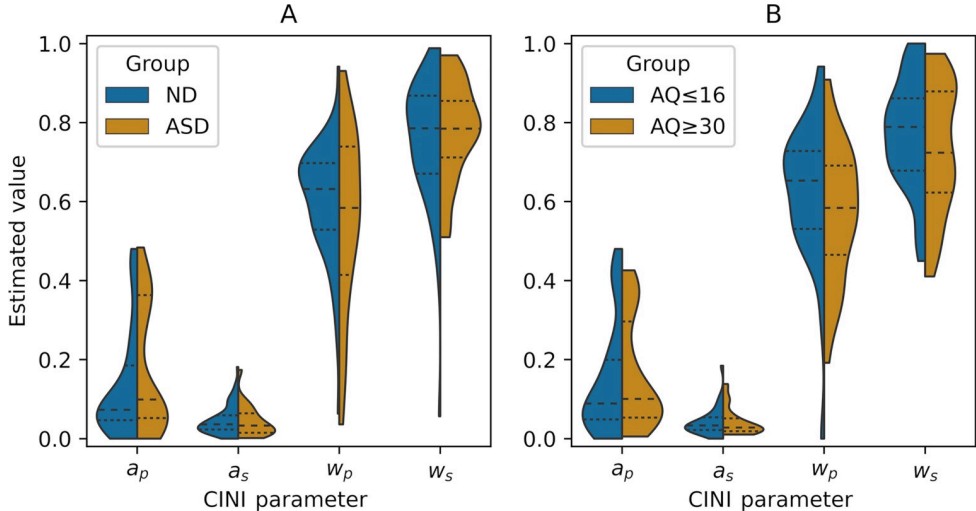

**Fig 6. CINI parameter values of ND vs ASD groups (A) and low-AQ vs high-AQ groups (B).** *Violin plots show the density of estimated parameters over the possible values, relative to the subgroup size. Dashed lines in the middle represent the median, while dotted ones represent the top and bottom quartiles in each group. No differences are observed between groups.*

separately. In schizophrenia, then, this imbalance would be larger and extend across the cognitive hierarchy, which would lead to interference between priors and likelihoods, making CII the better fit for these participants.

Surprisingly, we found no evidence for an association between prior or likelihood weights and ASD diagnoses or AQ scores. This result is seemingly in contrast with previous studies showing an overweighting of likelihoods relative to priors in autistic individuals or those with stronger autistic traits (e.g., [55,57–59]). However, these effects have been demonstrated exclusively in perceptual tasks, with the rare study of Bayesian decision-making in ASD showing no such imbalance [60,61]. Another important difference is that in most of the literature, participants have to learn prior beliefs based on the observed statistics, while in our study they are explicitly presented by the size of the baskets. It is possible that the cause of the prior-likelihood imbalance found in the literature lies in impaired prior acquisition, rather than in the relative weighting of the prior per se.

Our analysis revealed a slight increase of absolute confidence with stronger non-clinical delusional beliefs (PDI), but no association between PDI and any reverberation parameter.

**Table 3. Mann-Whitney U test results between the CINI parameters of the ASD and ND groups and the low-AQ and high-AQ groups.**

| CINI params | ND vs ASD | | | low-AQ vs high-AQ | | |
|---|---|---|---|---|---|---|
| | *f* | *p* | BF$_{01}$ | *F* | *p* | BF$_{01}$ |
| $a_p$ | 0.55 | 0.50 | 3.82 | 0.54 | 0.63 | 3.60 |
| $a_s$ | 0.45 | 0.46 | 3.15 | 0.47 | 0.70 | 3.88 |
| $w_p$ | 0.47 | 0.64 | 3.85 | 0.40 | 0.19 | 1.96 |
| $w_s$ | 0.53 | 0.71 | 3.71 | 0.42 | 0.31 | 2.17 |

Total AQ scores were used for the comparisons. f signifies the common language effect size, with larger f values corresponding to larger parameter values for the ASD and the high-AQ groups. An f of 0.5 corresponds to no differences. p-values are not adjusted for multiple comparisons. BF$_{01}$ stands for the Bayes factor 01, with higher values corresponding to stronger evidence for the null hypothesis.

This confirms the Jardri et al. findings of no such relationship in healthy subjects, although only 8 participants had scores above the clinical PDI mean of 11.9 [42]. This result would deserve further investigation with a more thorough assessment of schizotypy, so as to assess how it can fit with the dimensional view of schizophrenia [62]. Interestingly, PDI scores showed a significant interaction with likelihoods in the linear mixed-effects models and a slight correlation with the likelihood weights. While the latter result was not significant when adjusted for multiple comparisons, both of them agree with the prominent theory of over-weighted likelihoods in schizophrenia (e.g., [16]).

## Limitations and future work

Through our recruitment methods, we had aimed to recruit participants with a broad range of autistic traits. However, the resulting variance of AQ in our sample (SD 6.5) was only marginally higher than what is found in the general population (SD 5.6, [43]). Moreover, only 4 participants had an AQ score of more than 1 SD below the neurotypical mean of 16.9 [43] and only 5 participants had an AQ above the clinical mean of 35.2 [43]. A wider range of autistic traits would be useful in investigating Bayesian impairments that might be associated with the extremes of the AQ distribution. Moreover, the diagnoses of our participants in the Prolific subsample were self-reported. What those diagnoses were based on, and when they were delivered is uncertain, which could explain the atypical AQ scores of the ASD group. Our findings will need to be confirmed in a sample verified by a mental health professional, especially as the criteria for an ASD diagnosis have largely changed between versions of the Diagnostic and Statistical Manual of Mental Disorders [63]. Such a study would also benefit from cognitive measures, to ensure that perceptual or verbal reasoning abilities do not constitute a confounder for any differences between the groups. Another limitation that also nuances the comparison with previous investigations in schizophrenia [27] concerns the fact that our experiment took place online. The lack of a lab-controlled environment could have substantially affected the quality of the collected data. Adding to that is the absence of in-person communication between participants and researchers, so the instructions of the task could have been clearly conveyed and possible questions answered. Such effects were visible in our dataset by the large portion of subjects that were excluded ($\approx$14%).

In the fisher task, the baskets are presented before the lakes. This means that participants might simply display a recency bias, where the most recent evidence is overweighted. Under the Bayesian framework, the earlier evidence should create a prior belief in the participants, which is then combined with and updated by following evidence. Therefore, a recency bias is indistinguishable from an overweighting of sensory evidence. A possible issue, though, is that behavioural differences might be related to differences in the working memory of the participants. This could be especially important, since working memory is impaired in both ASD and schizophrenia [64,65]. However, Jardri et al. measured working memory performance in their sample and showed that it is correlated only with the prior weights, but not with the reverberation parameters. This would need to be validated in further studies, but we therefore expect that our findings regarding circular inference in autism should be robust to potential differences in working memory.

As with other findings relating behaviour to Bayesian inference impairments, it will be important to assess how our findings can be generalised to other tasks or modalities. Circular inference is formalised within a hierarchical Bayesian framework of cognitive processing. This framework assumes that priors express the (top-down) influences of the more abstract representations of the environment to the less abstract ones, while likelihoods encode the reverse (bottom-up) influences [66]. It is difficult to verify that the information presented in the current task (baskets and fish proportions) is encoded by subjects in the expected way–that is, that

the preferences of the fisherman correspond to more abstract or contextual information and the fish proportions to more sensory. If these stimuli were processed by the participants as being in the same conceptual level, the task structure would be more akin to a delayed cue integration task [67]. Additionally, it is possible that the basket size is treated by some participants as a qualitative variable, leading them to disregard the exact difference in size, something that would appear as prior overcounting in the models (although see Section D4 in S1 Supplementary Information). We believe that these concerns do not invalidate our results, but further research would be needed to understand how delayed cue integration tasks or qualitative information fit within the circular inference framework.

Future research should replicate both ASD and SCZ findings in other tasks, involving different cognitive modalities. The social beads task of Simonsen et al. [29], for example, might be well suited for the investigation of signal reverberation in ASD, given the condition's impairments. Perceptual tasks, on the other hand, would avoid conscious strategies that are especially prevalent in decision-making, focusing instead on more fundamental computations in the brain and connecting circular inference with the rest of the Bayesian literature. Equally important is clarifying the connection between reverberation and neurophysiological measures, with a focus on the spatial patterns of E/I imbalances across brain areas. Differences in such patterns could explain why computational [14,15] and neurobiological [4,9] theories of ASD and SCZ partially overlap, while their phenotypic expressions differ [53].

## Supporting information

**S1 Supplementary Information. Additional analyses and visualisations.** Table B3 in S1 Supplementary Information. Kendall rank correlations between recovered CINI parameters. Table B4 in S1 Supplementary Information. Kendall rank correlations between recovered CII parameters. Fig D1 in S1 Supplementary Information. CINI (top) and CII (bottom) model fit vs participant logit confidence estimates. Fig E1 in S1 Supplementary Information. Reverberation parameters of the current study's ASD sample and Jardri et al.'s SCZ sample. Table E3 in S1 Supplementary Information. Fixed and random effects model comparisons in both studies. (PDF)

## Author Contributions

**Conceptualization:** Nikitas Angeletos Chrysaitis, Renaud Jardri, Sophie Denève, Peggy Seriès.

**Formal analysis:** Nikitas Angeletos Chrysaitis, Peggy Seriès.

**Methodology:** Nikitas Angeletos Chrysaitis, Renaud Jardri, Sophie Denève, Peggy Seriès.

**Project administration:** Peggy Seriès.

**Software:** Nikitas Angeletos Chrysaitis, Renaud Jardri, Sophie Denève.

**Supervision:** Renaud Jardri, Sophie Denève, Peggy Seriès.

**Visualization:** Nikitas Angeletos Chrysaitis.

**Writing – original draft:** Nikitas Angeletos Chrysaitis.

**Writing – review & editing:** Renaud Jardri, Sophie Denève, Peggy Seriès.

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
