## [Decision Letter · Decision Letter 0]

2 Jun 2021

Dear Dr. Series,

Thank you very much for submitting your manuscript "No increased circular inference in autism or autistic traits" for consideration at PLOS Computational Biology.

As with all papers reviewed by the journal, your manuscript was reviewed by members of the editorial board and by several independent reviewers. In light of the reviews (below this email), we would like to invite the resubmission of a significantly-revised version that takes into account the reviewers' comments. As you can see from the reviewers' comments, there were major concerns about the self-declaration of the diagnoses, the parameter recoverability, and the interpretation of priors. Please address all of the raised points in full.

We cannot make any decision about publication until we have seen the revised manuscript and your response to the reviewers' comments. Your revised manuscript is also likely to be sent to reviewers for further evaluation.

Sincerely,

Tobias U Hauser, PhD

Associate Editor

PLOS Computational Biology

Samuel Gershman

Deputy Editor

PLOS Computational Biology

Reviewer's Responses to Questions

**Comments to the Authors:**

Reviewer #1: Jardri & Denève (2013, Brain) proposed the circular inference theory—defective Bayesian inference in the brain—as a unifying account to explain many different symptoms in schizophrenia and autism. Computational models based on the circular inference theory had been tested in previous studies for schizophrenia patients versus healthy controls, whose results provide strong support for the theory. Using similar computational models, the current study provides the first empirical test of the circular inference theory on autism and autistic traits. They find that autism or autistic traits, different from schizophrenia, have little influences on human subjects’ circular inference tendency.

The technical quality of the present study is high and the paper is beautifully written. Though some of its results seem to contradict those of a previous study (Jardri et al., 2017, Nature Communications) that used the same task and a similar design (i.e., whether the data were better fit by the CII or CINI model), the authors’ supplemental data analyses suggest that these differences can partly be explained by the slightly different experimental designs in the two studies.

The only issue that has raised my concern is the parameter recovery results reported in the Supplement for the two circular inference models (CINI and CII) . As Figs. S2 and S3 show, when the simulated a_p is around 0.1, the recovered a_p does not follow a unimodal distribution centered at 0.1, but follows a bimodal distribution peaked at 0.1 and 0.4. Similarly, when the simulated a_p is 0.4, the recovered a_p follows a bimodal distribution peaked at 0.1 and 0.4. The situation is similar for a_s. It occurs to me that 0.1 and 0.4 may be two “attractors” for the estimated values of a_p and a_s. In other words, the values of a_p and a_s in the circular inference models may not be reliably estimated, at least from the current trial set.

I am wondering whether the same bimodal-attractor issue would have occurred when the full set of 200 trials in Jardri et al., 2017 were used for parameter estimation. That is, whether the issue was closely associated with the circular inference models themselves, or only occurred for the subset of trials used in the present study. Is it possible for the authors to report the results of a parameter recovery analysis based on the fuIl set of 200 trials? I understand that the results of this additional analysis are unlikely to change the conclusion, but would appreciate it given the light it may shed on future studies.

Minor:

1. One relevant study: Lu, Yi, and Zhang (2019, PLoS CB) investigated the influence of autistic traits on information sampling behaviors in the framework of Bayesian inference. Different from the findings of previous perceptual inference studies, they found little evidence that people with different autistic traits would differ in their weighting of likelihoods.

Lu, H., Yi, L., & Zhang, H. (2019). Autistic traits influence the strategic diversity of information sampling: Insights from two-stage decision models. PLoS Computational Biology, 15(12), e1006964. doi:10.1371/journal.pcbi.1006964

2. What are the neurobiological and psychological implications for the differences between the CII and CINI models?

Reviewer #2: the review is uploaded as an attachment

Reviewer #3: Summary

This paper extends a previously used empirical/computational paradigm for circular inference, previously used to account for the impact of aberrant excitation/inhibition on the integration of prior knowledge with sensory information in schizophrenia (scz), to autism (asd). The extension of the methods of the scz study to asd is motivated by evidence of aberrant e/i in both conditions and commonalities in computational accounts of these. The study was conducted via an online platform and the sample included individuals from the general population (and autism quotient AQ as a measure of autism symptomatology) as well as autistic individuals (who, intriguingly, did not score as high as expected in AQ). There were no systematic differences in task performance associated with autism traits in the general population or with an autism diagnosis - in either the model-free or the

General comments

I think that this manuscript is a valuable contribution to the fields of bayesian accounts of autism and, more broadly, computational psychiatry. The empirical and computational methods were presented clearly and the code was documented in detail.

I would like to raise the following general comments about the approach:

The paper draws on earlier computational research in perception in autism (references [55-58]) and suggests that the contrast between the current data and these studies could be related to the use of decision-making rather than perceptual tasks. In my opinion, two points need to be considered

The most important difference is that in some of these studies limitations related to prior knowledge could be due to aberrant mechanisms in extracting prior knowledge statistics. In this task, prior knowledge is not formed across trials, but is instead is presented in each trial. It is as if participants are asked to mentalise, “suppose you have this knowledge. What would you decide?”- rather than experience the knowledge themselves.

There is a substantial perception element (size or numerosity) influencing the interpretation of priors and likelihoods with baskets and ponds.

The unexpected patterns of not-so-high AQ in the autistic group is puzzling and possibly worrying. Overall, the authors provide good detail on how they mitigated potential limitations of requirement and testing via online platforms (I suspect as an alternative given covid-19?). For example, it was useful to include attentional controls in the task. Similarly, I think the authors needed to include a second measure of autistic symptomatology, e.g., SRS-2. to corroborate the autism diagnosis. This is deemed as necessary in studies that involve face-to-face testing, and more so it would be very important in an online setting.

Corsello C, Hus V, Pickles A, Risi S, Cook EH Jr, Leventhal BL, Lord C. Between a ROC and a hard place: decision making and making decisions about using the SCQ. J Child Psychol Psychiatry. 2007 Sep;48(9):932-40. doi: 10.1111/j.1469-7610.2007.01762.x. PMID: 17714378.

Another major limitation is the lack of cognitive measures. In the original study on scz, the two groups were closely matched with each other in neuropsychological evaluation measures. I think having a measure of performance and verbal intelligence was relevant here. In terms of methodological rigour, this could ensure that any potential group differences are not due to differences in perceptual or verbal reasoning abilities, but related to autism symptomatology. These measures are important as this task relies on perceptual and verbal reasoning abilities.

I think that the authors need to make a clearer decision on whether they situate this study as a cross-syndrome comparison of asd and scz with neurotypicals as reference or whether they compare asd to nt with reference to a relevant earlier scz study. If the former, I think that a direct comparison with the findings with the scz individuals is missing. There was a stark contrast between scz and nt performance in the original study, and no difference at all in this study. This message should be conveyed more clearly. Some considerations are given in supplementary info, a plot would be needed too.

Minor comments:

p.8

Interestingly, the participants with the ASD diagnoses had AQ scores on the low-end (M=28.0, 146 SD=8.0) compared to those reported in the literature for autistic individuals (M=35.2, SD=6.3) [43]

- What was the gender distribution in the autism group?

What do the authors make of it? Is it limitations of AQ or limitations of online recruitment or chance? This needs to be extended.

-line 275 to outliers [50] (Fig A2 and A3 in S1 Supplementary Information).S2 S3?

-and line 277 also showed no correlation between different parameters (Table A3 and A4 in S1...: It is S3 and S4?

- I think that a discussion paragraph outlining the similarities of accounts of autism and schizophrenia would be a useful addition.

**Have the authors made all data and (if applicable) computational code underlying the findings in their manuscript fully available?**

Reviewer #1: Yes

Reviewer #2: None

Reviewer #3: Yes

PLOS authors have the option to publish the peer review history of their article (what does this mean?). If published, this will include your full peer review and any attached files.

Reviewer #1: No

Reviewer #2: No

Reviewer #3: No
---

## [Decision Letter · Decision Letter 1]

24 Aug 2021

Dear Dr. Series,

We are pleased to inform you that your manuscript 'No increased circular inference in adults with high level of autistic traits or autism' has been provisionally accepted for publication in PLOS Computational Biology.

Best regards,

Tobias U Hauser, PhD

Associate Editor

PLOS Computational Biology

Samuel Gershman

Deputy Editor

PLOS Computational Biology

Reviewer's Responses to Questions

**Comments to the Authors:**

Reviewer #1: The authors have satisfactorily addressed all my concerns.

Reviewer #2: Thank you for your excellent responses to my concerns. They have all been addressed sufficiently and I have no further comments.

Reviewer #3: I would like to thank the authors for their comprehensive response to my review. I think that all the comments raised by me and the other reviewers have been addressed satisfactorily in the revision.

**Have the authors made all data and (if applicable) computational code underlying the findings in their manuscript fully available?**

Reviewer #1: Yes

Reviewer #2: Yes

Reviewer #3: Yes

PLOS authors have the option to publish the peer review history of their article (what does this mean?). If published, this will include your full peer review and any attached files.

Reviewer #1: No

Reviewer #2: No

Reviewer #3: **Yes: **Themis Karaminis

---

## [Editor Report · Acceptance letter]

17 Sep 2021

PCOMPBIOL-D-21-00707R1 

No increased circular inference in adults with high levels of autistic traits or autism

Dear Dr Seriès,

I am pleased to inform you that your manuscript has been formally accepted for publication in PLOS Computational Biology. Your manuscript is now with our production department and you will be notified of the publication date in due course.

With kind regards,

Katalin Szabo
